# Search or Accelerate: Confidence-Switched Position Beam Search for Diffusion Language Models

Mingyu Cao[1]  Alvaro H.C. Correia[2]  Christos Louizos[2]  Shiwei Liu[3 4 5] †  Lu Yin[1] †

## Abstract

Diffusion Language Models (DLMs) generate text by iteratively denoising a masked sequence, repeatedly deciding *which positions* to commit at each step. Standard decoding follows a greedy rule, unmasking the most confident positions, yet this local choice can lock the model into a suboptimal unmasking order, especially on reasoning-heavy prompts. We present **S**earch **O**r **A**ccele**R**ate (SOAR), a **training-free** decoding algorithm that adapts its behavior to the model's uncertainty. When confidence is low, SOAR briefly widens the search over alternative unmasking decisions to avoid premature commitments; when confidence is high, it collapses the search and decodes many positions in parallel to reduce the number of denoising iterations. Across mathematical reasoning and code generation benchmarks (GSM8K, MBPP, HumanEval) on DREAM-7B and LLADA-8B, SOAR improves generation quality while maintaining competitive inference speed, offering a practical way to balance quality and efficiency in DLM decoding. The code is available at https://github.com/duterscmy/SOAR

## 1. Introduction

Diffusion Language Models (DLMs) (Ye et al., 2025; Nie et al., 2026; Zhu et al., 2025a) have recently emerged as a promising alternative to autoregressive (AR) generation (Touvron et al., 2023; Yang et al., 2025a). Unlike AR models that decode tokens sequentially in a fixed left-to-right order, DLMs iteratively refine a partially masked sequence and can decode multiple token positions in parallel. In practice,

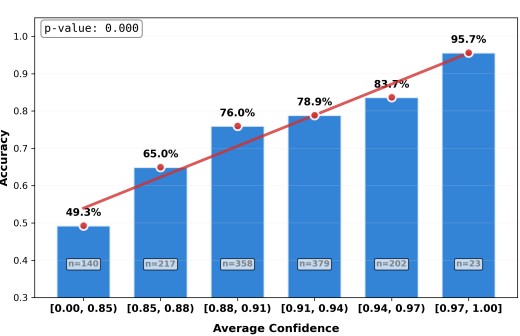

*Figure 1.* Accuracy and average confidence on GSM8K with DREAM-7B-Base. All questions are divided into 6 bins by their average decoding confidence, with $n$ indicating the sample size in each bin. The red solid line represents the trend line.

most mask-based DLM decoders follow a simple greedy rule: at each denoising step, they unmask the positions with the highest prediction confidence and keep the remaining positions as [MASK].

This greedy unmasking rule is attractive for efficiency, but it also makes a strong local decision about *which positions to commit next*. Importantly for DLMs, this unmasking order is not fixed (unlike left-to-right AR decoding), and different unmasking schedules can lead to different final generations. This motivates our central question: **Can we improve DLM decoding by explicitly exploring alternative unmasking orders beyond greedy selection?**

A natural signal for guiding such exploration is the model's prediction confidence, which has been leveraged in prior work for adaptive-parallel decoding (Chen et al., 2025). In our setting, we observe the same qualitative pattern: on GSM8K (Cobbe et al., 2021), examples decoded with higher average token confidence tend to be more accurate (Figure 1). This suggests that searching for decoding trajectories that maintain higher confidence may improve downstream quality, especially on reasoning-heavy prompts.

Inspired by the success of beam search in AR decoding (Freitag & Al-Onaizan, 2017), we adapt this paradigm to DLMs. A key difference is that, while AR beam search expands candidates primarily in the token/vocabulary space,

[1]University of Surrey [2]Qualcomm AI Research. Qualcomm AI Research is an initiative of Qualcomm Technologies, Inc. [3]ELLIS Institute Tübingen [4]Max Planck Institute for Intelligent Systems [5]Tübingen AI Center. Correspondence to: Lu Yin <l.yin@surrey.ac.uk>, Shiwei Liu <sliu@tue.ellis.eu>.

*Proceedings of the 43rd International Conference on Machine Learning*, Seoul, South Korea. PMLR 306, 2026. Copyright 2026 by the author(s).

DLM decoding admits a distinct search dimension: the *position space*, i.e., the order in which masked positions are unmasked. We find that extending the search to the position space consistently improves output quality across benchmarks, but it also increases computation roughly linearly with beam width. This leads to a practical challenge: **How can we retain the gains from position-space search without incurring prohibitive inference latency?**

To address this, we propose SOAR, a **training-free** confidence-switched position beam search procedure that dynamically balances exploration and speed based on the model's confidence at each step. When the model is uncertain, SOAR widens the beam to explore multiple unmasking choices and avoid premature commitments; when the model is confident, it decodes many high-confidence positions in parallel and collapses the beam to accelerate inference. This enables the model to spend compute when needed and decode quickly when possible, yielding a practical quality–speed trade-off for DLM inference.

Our contributions are summarized as follows:

- We introduce **P**osition **B**eam **S**earch (PBS), a training-free decoding method that explores alternative unmasking sequences in the position space and consistently improves output quality across multiple benchmarks.

- We propose SOAR, a training-free adaptive inference algorithm that switches between parallel decoding and position-space search based on per-step confidence, effectively balancing speed and quality.

- We demonstrate that SOAR integrates naturally with variable-length DLM decoding methods and multiple unmasking metrics, highlighting its flexibility and compatibility with advanced decoding strategies.

**Conflict of Interest Disclosure.** The authors declare no financial conflicts of interest related to this work.

## 2. Related Works

### 2.1. Diffusion Language Models

Diffusion Language Models have emerged as a competitive alternative to autoregressive models for text generation (Nie et al., 2026; Ye et al., 2025; Gong et al., 2025a). Unlike their autoregressive counterparts (Touvron et al., 2023; Yang et al., 2025a) that generate tokens sequentially, DLMs define a forward process that gradually corrupts text with noise (or [MASK] tokens) and learn a reverse process to reconstruct the original text through iterative denoising.

The development of large-scale DLMs has progressed along several distinct architectural paths. Two prominent approaches have emerged based on their initialization strate-

gies: methods that leverage pre-trained autoregressive models and those trained from scratch. DiffuLLaMA (Gong et al., 2025a) and DREAM (Ye et al., 2025) follow a transfer learning paradigm, where DLMs are initialized from existing strong autoregressive models (LLaMA (Touvron et al., 2023) and Qwen (Yang et al., 2025a), respectively). Their success demonstrates the viability of this approach, showing that DLMs can achieve competitive performance relative to strong autoregressive baselines. In contrast, LLADA (Nie et al., 2026) represents a distinct path, being trained from scratch as a high-performing open-source DLM. Together, these works have established DLMs as a credible and high-performance architecture for text generation. The standard decoding process for these mask-based DLMs is iterative and parallel. Starting from a fully masked sequence, the model predicts all tokens simultaneously at each denoising step. A common strategy involves selecting and committing the tokens at the most confident positions, while replacing the remaining uncertain positions with [MASK] tokens for the next round of refinement. This "predict-all, then commit-a-subset" loop repeats until the entire sequence is unmasked, leveraging the model's bidirectional context at every step to refine predictions.

### 2.2. Optimized Inference Methods for DLMs

The non-autoregressive, iterative nature of DLM decoding presents unique inference challenges, as the random access pattern across denoising steps invalidates the traditional KV-cache and necessitates pre-defining a maximum sequence length, often leading to computational redundancy. Consequently, research has focused on specialized acceleration techniques. One primary direction involves designing specialized KV-cache mechanisms that either leverage the high similarity of hidden states across steps for approximate caching or restructure decoding into a block-autoregressive manner to enable state reuse from previous context (Wu et al., 2025; Nguyen-Tri et al., 2025; Huang et al., 2025a; Liu et al., 2025; Ma et al., 2025). Another critical direction aims to reduce the total number of steps through parallel and adaptive-length decoding. This includes confidence-based strategies that dynamically determine how many tokens to decode in parallel, as well as early-commit methods which halt decoding once the model's predictions stabilize, significantly speeding up inference (Wu et al., 2025; Yang et al., 2025b; Gao et al., 2025; Li et al., 2025b;a). In addition, some works address the inherent inability of standard DLMs to revise committed tokens, exploring solutions ranging from training-free correctors to modifications of the diffusion process itself (Huang et al., 2025b; Mounier & Idehpour, 2025; Zhu et al., 2025b). A concurrent work, Order-Token Search (Shen et al., 2026), proposes to simultaneously explore the decoding space across both the position and token dimensions to achieve better results, but at the

expense of inference speed. Our proposed method, SOAR, dynamically switches between search and parallel decoding modes based on the model's confidence during token decoding. This enables consistent improvements across multiple benchmarks without sacrificing decoding speed.

## 3. Methodology

### 3.1. Preliminaries of DLM Inference

Given a text sequence of length $L$, we denote $x = [x_1, x_2, \ldots, x_L] \in \mathcal{V}^L$ where $\mathcal{V}$ is the vocabulary. In diffusion language models, the forward process gradually corrupts $x$ through $T$ steps:

$$q(x_t|x_{t-1}) = \begin{cases} \mathcal{M}(x_{t-1}, \rho_t) & \text{if } t > 0 \\ \delta(x) & \text{if } t = 0 \end{cases} \quad (1)$$

where $\mathcal{M}(\cdot, \rho_t)$ masks a proportion $\rho_t$ of tokens, and $\delta(\cdot)$ is the Dirac delta function. The reverse process iteratively reconstructs the original text:

$$p_\theta(x_{t-1}|x_t) = \prod_{i=1}^{L} p_\theta(x_{t-1,i}|x_t) \quad (2)$$

where $x_{t,i}$ denotes the $i$-th token at step $t$, and $p_\theta$ is parameterized by the DLM.

At inference time, starting from a fully masked sequence $x_T = [[\text{MASK}]]^L$, the model generates text through $T$ denoising iterations. At each step $t$, the model predicts a distribution over $\mathcal{V}$ for every position $i$:

$$\mathbf{p}_{t,i} = \text{softmax}(f_\theta(x_t)_i) \in \mathbb{R}^{|\mathcal{V}|} \quad (3)$$

where $f_\theta$ is the DLM. The decoding strategy determines which tokens to decode at each step.

Let $\mathcal{I}_t \subseteq \{1, \ldots, L\}$ be the set of positions selected for unmasking at step $t$. The standard approach selects $k$ positions with the highest confidence scores:

$$\mathcal{I}_t = \text{topk}\left(\{\max_{v \in \mathcal{V}} \mathbf{p}_{t,i}[v] : i \in M_t\}, k\right) \quad (4)$$

where $M_t = \{i : x_{t,i} = [\text{MASK}]\}$ represents the set of masked positions. Then the updated sequence is:

$$x_{t-1,i} = \begin{cases} \arg\max_{v \in \mathcal{V}} \mathbf{p}_{t,i}[v] & \text{if } i \in \mathcal{I}_t \\ [\text{MASK}] & \text{otherwise} \end{cases} \quad (5)$$

The key challenge is how to select $\mathcal{I}_t$ to maximize sequence quality while minimizing computational cost.

### 3.2. Position Beam Search (PBS)

However, the overall confidence of sequences generated through greedy decoding may not be optimal. To obtain decoding paths with higher confidence, we introduce a training-free method that adapts the classic beam-search paradigm to a new dimension—the position space. PBS is initialized with a single sequence consisting only of masked tokens and a score of zero: $\mathcal{B}_0 = \{(x_0, 0)\}$, where $x_0$ contains only [MASK] tokens.

Let $\mathcal{B}_t = \{(x_t^{(1)}, s_t^{(1)}), (x_t^{(2)}, s_t^{(2)}), \ldots, (x_t^{(K)}, s_t^{(K)})\}$ be defined as the beam of size $K$ at step $t$, where each $x_t^{(j)}$ is a candidate sequence and $s_t^{(j)}$ is its cumulative score, with $j$ indexing the candidate in the beam.

#### 3.2.1. CANDIDATE GENERATION

For each candidate $(x_t^{(j)}, s_t^{(j)})$ in the beam, we generate new candidates by exploring different ways to unmask its currently masked tokens. Specifically, for each possible set of $n$ masked positions (denoted by $\mathcal{I}$) that we choose to unmask in this step, we obtain a new sequence $x_{t-1}^{(j)}$ and compute its cumulative score as:

$$s_t^{(j)} + \phi(\mathcal{I}, \mathbf{p}_t^{(j)}), \quad (6)$$

where $\mathbf{p}_t^{(j)}$ represents the predicted token distributions for candidate $j$, and $\phi(\mathcal{I}, \mathbf{p})$ is the average confidence score of the selected positions:

$$\phi(\mathcal{I}, \mathbf{p}) = \frac{1}{|\mathcal{I}|} \sum_{i \in \mathcal{I}} \max_{v \in \mathcal{V}} \mathbf{p}_i[v]. \quad (7)$$

Using the average instead of the sum in the scoring function aims to prevent sequences with more remaining masked tokens from being unfairly penalized.

#### 3.2.2. POSITION SELECTION STRATEGIES

Let $\mathcal{P}_t^{(j)}$ denote the set of candidate positions selected from $M_t^{(j)}$ for exploration. The positions are always ranked by confidence:

$$\mathcal{P}_t^{(j)} = \{i_1, i_2, \ldots, i_{|\mathcal{P}_t^{(j)}|}\},$$
$$\text{where } \max_v \mathbf{p}_{t,i_1}^{(j)}[v] \geq \max_v \mathbf{p}_{t,i_2}^{(j)}[v] \geq \cdots \quad (8)$$

**Case 1: Single-token Decoding** ($n = 1$)

When only one token is unmasked per step, which is a boundary case of parallel decoding, candidates are generated by selecting only one token from $\mathbf{p}_t^{(j)}$, and the tokens selected by different candidates are mutually exclusive.

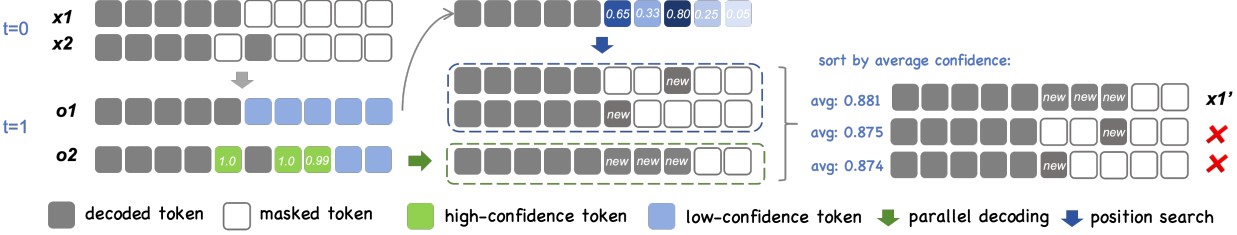

*Figure 2.* Overview of SOAR. When $t = 0$, there are two sequences in the beam. Based on the confidence of decoded tokens, one undergoes parallel decoding (green arrow), while the other undergoes position search (blue arrow). After reordering, since the sequence with the highest average confidence is obtained through parallel decoding, the beam size is reduced to 1.

**Case 2: Parallel Decoding** ($n > 1$)

When $n$ tokens are unmasked per step, firstly, the minimum number of candidate tokens $M$ required to generate $K$ distinct combinations is determined:

$$M = \min \left\{ m : \binom{m}{n} \geq K, m \leq |M_t^{(j)}| \right\} \quad (9)$$

If $|M_t^{(j)}| < M$, all masked positions are selected, so $M = |M_t^{(j)}|$. Then $\mathcal{P}_t^{(j)}$ contains the top-$M$ positions by confidence. To avoid the combinatorial explosion of generating all $\binom{M}{n}$ subsets, we efficiently select the top-$K$ $n$-element subsets from $\mathcal{P}_t^{(j)}$ by employing a best-first search strategy over the confidence-ordered positions, ensuring that only the most promising combinations are explored. These $K$ subsets are then used as candidate position sets for parallel unmasking.

### 3.2.3. BEAM UPDATE

All candidates from all beams are pooled, sorted by their scores, and the top $K$ form the new beam:

$$\mathcal{B}_{t-1} = \arg \max_{\mathcal{C} \subseteq \bigcup_{j=1}^{K} \mathcal{C}_t^{(j)}, |\mathcal{C}|=K} \sum_{(x,s) \in \mathcal{C}} s \quad (10)$$

Table 1 presents experimental results for single-token decoding and parallel decoding settings. In the single-token decoding setting, PBS consistently improves performance across multiple benchmarks. However, this improvement comes at a computational cost: computation scales linearly with beam size, resulting in proportionally slower decoding.

Enabling PBS with parallel decoding makes inference speed comparable to greedy decoding, yet leads to a noticeable drop in accuracy relative to the single-token-per-step setup. We hypothesize that this degradation stems from the confidence-agnostic nature of parallel decoding: by forcing the model to decode a fixed number of tokens at each step, it may be compelled to generate low-confidence tokens, thereby compromising output quality. We examine this hypothesis in more detail in Section 4.4.

To navigate this trade-off between decoding speed and output confidence, we introduce SOAR, a method that dynamically decides when to accelerate via parallel decoding and when to refine via confidence-based search.

### 3.3. SOAR: Confidence-Switched Position Beam Search

SOAR addresses this trade-off by dynamically adapting the decoding strategy according to prediction confidence, enabling the model to explore more when uncertain and decode faster when confident. Let $\mathcal{C}_t = \{(x_t^{(1)}, s_t^{(1)}), \ldots, (x_t^{(N)}, s_t^{(N)})\}$ denote the set of candidate sequences at step $t$, where $N_t = |\mathcal{C}_t|$ is the candidate set size, each $x_t^{(j)}$ represents a candidate sequence, and $s_t^{(j)}$ is its cumulative score, with $j$ indexing candidates in the beam.

#### 3.3.1. CONFIDENCE-GUIDED ADAPTIVE DECODING

At each decoding step $t$, for each candidate sequence $x_t^{(j)}$ in the candidate set, we compute confidence scores for all masked positions:

$$c_{t,i} = \max_{v \in \mathcal{V}} \mathbf{p}_{t,i}[v] \quad (11)$$

$x_t^{(j)}$ switches between two decoding modes based on the presence of high-confidence positions for each candidate sequence:

(a) **Parallel Decoding Mode**: When there exists at least one high-confidence position, SOAR directly decodes all such positions in parallel:

$$\mathcal{I}_t = \{i : c_{t,i} > \tau\} \quad (12)$$

All tokens in $\mathcal{I}_t$ are unmasked simultaneously, where $\tau$ is a hyper-parameter. This mode is similar to the design of Fast-dLLM (Wu et al., 2025), with the difference that when confidence falls below the threshold, SOAR will switch to beam search mode to explore other possible decoding paths.

(b) **Beam Search Mode**: When no masked position exceeds the confidence threshold ($c_{t,i} \leq \tau$ for all $i \in M_t^{(j)}$), SOAR falls back to position beam search with $k = 1$ token per step. The beam explores the top-$K$ most confident positions individually, trading speed for more thorough exploration.

### 3.3.2. DYNAMIC CANDIDATE SIZE ADJUSTMENT

Sequences in the candidate set generate new candidate sequences, which are then sorted in descending order based on their average decoding confidence. After sorting, only the top $N_t$ sequences are retained, where $N_t$ denotes the size of the candidate set at step $t$.

To further optimize efficiency, SOAR dynamically adjusts $N_t$ after each decoding step based on the origin of the new candidate with the highest score. The decision rule is as follows: if the candidate with the highest score originates from the parallel decoding mode, the model is considered sufficiently confident and $N_t$ is reduced to 1 for the next step; otherwise, if the best new candidate comes from the PBS mode, $N_t$ is reset to the preset size $K$, which corresponds to the maximum beam width used in the PBS mode. Formally,

$$N_t = \begin{cases} K & \text{if the best candidate is from PBS} \\ 1 & \text{if it is from parallel decoding.} \end{cases} \quad (13)$$

This adaptive strategy ensures that when the model exhibits high confidence, computational resources are concentrated on the most promising path; when uncertainty arises, multiple decoding hypotheses are retained to avoid local optima.

The time complexity of SOAR is $O(T \cdot \bar{N})$, where $T$ is the number of decoding steps, and $\bar{N}$ is the average candidate size. In practice, because a significant portion of the model's tokens have confidence above the threshold, $\bar{N}$ remains well below $K$, and the overall step count $T$ is also reduced.

SOAR thus provides a principled way to balance the exploration-exploitation tradeoff in DLM decoding, adapting to the model's uncertainty at each step to optimize both quality and efficiency. Appendix B presents a visual analysis of the algorithm's behavior.

## 4. Experiments

### 4.1. Experiment Setup

We conduct experiments on two representative diffusion language models: LLaDA-8B (Nie et al., 2026) and DREAM-7B (Ye et al., 2025), both using the Base version. To ensure reproducibility, all experiments are conducted with a temperature of 0, using softmax probabilities as confidence scores. Unless otherwise specified, the threshold $\tau = 0.95$ for LLaDA-8B and $\tau = 0.90$ for DREAM-7B and the maximum beam size $K$ is set to 2. The maximum sequence length is set to 256 and 512, respectively, to validate the

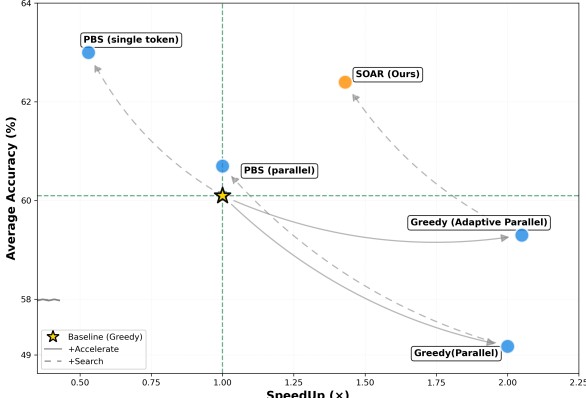

*Figure 3.* Pareto frontier on DREAM-7B-Base. Solid arrows indicate adding parallel decoding (acceleration), and dashed arrows indicate adding position beam search. This plot is generated using the average accuracy and average speedup from Table 1. For better visual presentation, the Y-axis range 49–58 is compressed.

robustness of the results.

To comprehensively assess the effectiveness of our approach, experiments are conducted across three benchmarks covering code generation and mathematical reasoning. For mathematical reasoning, we employ GSM8K (Cobbe et al., 2021), a dataset of grade-school math word problems. For code generation, we evaluate on MBPP (Austin et al., 2021), which contains entry-level Python programming tasks, and HumanEval (Chen et al., 2021), a collection of handwritten programming challenges designed for synthesis evaluation.

We adhere to standard few-shot evaluation protocols for each benchmark: 0-shot for HumanEval, 3-shot for MBPP and 4-shot for GSM8K. Accuracy is used as the evaluation metric for mathematical reasoning, while pass@1 is adopted for code generation benchmarks. All experiments are conducted based on open-source lm-evaluation-harness (Gao et al., 2024) with a single NVIDIA A100 80GB GPU. The total inference time of the benchmark is used to calculate the SpeedUp ratio compared with standard greedy decoding.

### 4.2. Main Results on Accuracy and Speed

Table 1 summarizes the performance of different decoding methods across both models and three benchmarks. In addition, we plot the variation of average accuracy and SpeedUp among these decoding methods in Figure 3 for better illustration. In most settings, PBS consistently improves accuracy over greedy decoding, with DREAM-7B achieving gains of up to $+7.3\%$ on HumanEval. However, this quality improvement comes at the cost of nearly doubled inference latency (average SpeedUp $\times 0.54$), as exploring multiple decoding paths increases computation.

*Table 1.* Main Results. The upper section presents results on LLADA-8B-Base, while the lower section shows results on DREAM-7B-Base.

| Benchmark | HumanEval | | MBPP | | GSM8K | | Avg. |
|---|---|---|---|---|---|---|---|
| Max Length | 256 | 512 | 256 | 512 | 256 | 512 | |
| **Greedy** | 32.3 | 32.9 | 40.8 | 39.2 | 70.4 | 70.9 | 47.8 |
| **Greedy (Adaptive Parallel)** | 32.3 | 32.9 | 40.8 | 39.2 | 70.4 | 71.0 | 47.8 |
| *SpeedUp* | *×2.25* | *×2.79* | *×2.06* | *×2.42* | *×1.69* | *×1.92* | *×2.19* |
| **PBS (Single Token)** | 34.2 | 34.8 | 41.4 | 39.2 | 71.6 | 72.1 | 48.9 |
| *SpeedUp* | *×0.48* | *×0.48* | *×0.48* | *×0.48* | *×0.47* | *×0.48* | *×0.48* |
| **PBS (Parallel)** | 31.1 | 29.3 | 35.2 | 36.1 | 69.5 | 68.1 | 44.9 |
| *SpeedUp* | *×0.98* | *×0.98* | *×0.97* | *×0.97* | *×0.98* | *×0.99* | *×0.98* |
| **SOAR** | 32.9(+0.6) | 39.0(+6.1) | 40.8 | 39.4(+0.2) | 71.3(+0.9) | 71.5(+0.6) | 49.2(+1.4) |
| *SpeedUp* | *×1.63* | *×2.16* | *×1.49* | *×1.85* | *×1.18* | *×1.43* | *×1.62* |
| **Greedy** | 50.0 | 53.7 | 53.4 | 55.4 | 73.7 | 74.5 | 60.1 |
| **Greedy (Adaptive Parallel)** | 50.0 | 51.8 | 53.8 | 55.6 | 73.2 | 74.6 | 59.8 |
| *SpeedUp* | *×1.63* | *×1.60* | *×2.20* | *×2.86* | *×2.19* | *×1.83* | *×2.05* |
| **PBS (Single Token)** | 57.3 | 58.1 | 54.0 | 56.9 | 75.2 | 76.4 | 63.0 |
| *SpeedUp* | *×0.57* | *×0.52* | *×0.50* | *×0.52* | *×0.57* | *×0.56* | *×0.54* |
| **PBS (Parallel)** | 53.7 | 54.3 | 53.6 | 54.0 | 73.5 | 75.1 | 60.7 |
| *SpeedUp* | *×1.02* | *×0.98* | *×1.01* | *×0.96* | *×1.03* | *×1.01* | *×1.00* |
| **SOAR** | 55.5(+5.5) | 55.1(+1.4) | 57.0(+3.6) | 56.2(+0.8) | 74.7(+1.0) | 75.7(+1.2) | 62.4(+2.3) |
| *SpeedUp* | *×1.13* | *×1.07* | *×1.87* | *×2.47* | *×0.95* | *×1.07* | *×1.43* |

PBS with parallel token decoding ($n = 2$) restores decoding speed to the same level as greedy decoding, yet the average accuracy also decreases significantly compared to PBS with single token. We attribute this to the fact that forcing a fixed number of tokens to be decoded in parallel can undermine the confidence-based selection that underlies PBS's effectiveness. In contrast, SOAR switches between search and parallel decoding based on confidence, avoiding excessive search that affects decoding speed while also preventing mandatory parallel decoding from harming global confidence, thereby achieving a balance between speed and quality improvement.

### 4.3. Ablation Study

**Confidence Threshold.** We examine how the confidence threshold $\tau$ affects decoding quality and speed. Figure 4(a) plots the cumulative distribution of token confidence scores on GSM8K under standard decoding, showing a long tail—about 50% of tokens have probability $> 0.95$. We vary $\tau \in [0.8, 0.98]$ and evaluate accuracy and SpeedUp with coding (MBPP) and math (GSM8K) tasks on DREAM-7B. As shown in Figure 4(b)-(c), raising $\tau$ from 0.8 to 0.98 steadily improves accuracy but reduces decoding speed. Notably, for $\tau > 0.8$, SOAR consistently beats standard decoding in accuracy while matching or exceeding its speed, demonstrating robustness. We conducted the same analysis

on LLADA-8B, and ultimately set $\tau = 0.95$ for LLADA-8B and $\tau = 0.9$ for DREAM-7B.

**Beam Size.** We further investigate whether expanding the beam size can yield additional performance gains. Figure 5 illustrates the trade-off between generation quality and inference speed under different beam sizes (ranging from 1 to 4) on the HumanEval benchmark using the DREAM-7B model. Setting beam size to 1 corresponds to using only confidence-based parallel decoding. While increasing the beam size may offer modest improvements in accuracy, the computational overhead grows linearly with beam width, resulting in significant slowdowns in inference speed. Therefore, we select a beam size of 2 as the default setting to balance quality and efficiency in our experiments.

### 4.4. Analysis of Confidence and AR-ness

**Confidence of Decoding Sequence.** To test the hypothesis proposed in Section 1—*Can we improve decoding quality by exploring alternative unmasking sequences with higher confidence beyond greedy selection?*—we analyze the average confidence of decoded sequences.

**Definition 4.1.** Average Confidence: For each sample, we extract the reasoning tokens that precede the answer keyword ("answer" token in GSM8K). Each token's confidence score is defined as the model's maximum softmax probability. The sample-level average confidence is computed

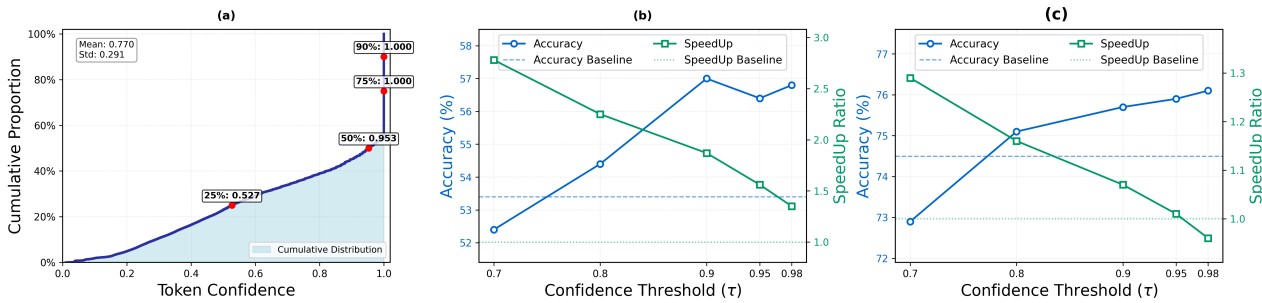

*Figure 4.* The threshold study on DREAM-7B: (a) Cumulative distribution of token confidence scores on GSM8K; (b) Trade-off between accuracy and SpeedUp under varying confidence thresholds on MBPP; (c) Trade-off between accuracy and SpeedUp under varying confidence thresholds on GSM8K.

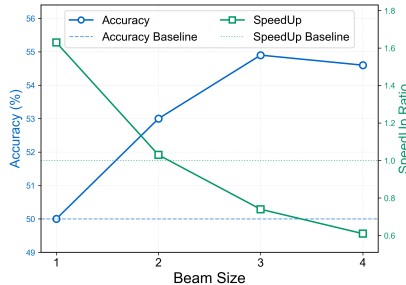

*Figure 5.* The beam size study on DREAM-7B: Trade-off between accuracy and SpeedUp under increasing beam size on HumanEval.

as the mean of these token-wise scores. We then average across all samples for each decoding method to obtain the method-level average confidence.

Figure 6(a) illustrates the relationship between average confidence and accuracy for various decoding methods on the GSM8K benchmark using DREAM-7B. Here, "Parallel" refers to decoding exactly two tokens per step, "Adaptive Parallel" decides whether to perform parallel decoding based on confidence thresholds, and the maximum sequence length is 512.

Our results show a positive correlation between accuracy and average confidence. PBS achieves an average accuracy 1.9% higher than Greedy Search, confirming our hypothesis. In contrast, Parallel decoding forces the generation of two tokens per step regardless of confidence, which significantly lowers average confidence and leads to a notable accuracy drop. When Parallel decoding is combined with PBS, the confidence gains from PBS are negated by the confidence loss from Parallel decoding, resulting in a 1.3% accuracy reduction compared to PBS alone. SOAR, however, dynamically switches between Parallel decoding and PBS based on confidence, preserving higher average confidence and maintaining competitive accuracy.

**AR-ness of Decoding Sequence.** We further investigate why SOAR yields better decoding sequences using another metric: the Global AR-ness proposed by (Gong et al., 2025b), which quantifies how much the unmasking schedule of a diffusion model resembles an autoregressive pattern, specifically, whether it follows a "left-first" pattern.

**Definition 4.2.** Global AR-ness: At each decoding step $t$, we examine whether the predicted token lies within the first $k$ masked positions. The Global AR-ness@$k$ is defined as the average ratio of such steps across the entire decoding process, measuring the tendency to always unmask the earliest remaining token and thus capturing a left-to-right filling strategy. This ratio increases with $k$, as the criterion becomes easier to satisfy when more early positions are allowed. *A higher value indicates that the generation is more autoregressive.*

We set $k = 5$ and compute the mean Global AR-ness across all samples. Figure 6(b) illustrates the relationship between negative Global AR-ness and accuracy. We observe that methods with lower Global AR-ness, i.e., less left-to-right-biased unmasking schedules, tend to achieve higher accuracy. This suggests that the unmasking order itself is an important factor in DLM decoding quality. One possible interpretation is that strongly left-to-right unmasking may prematurely commit to positions near the prefix, where the model often assigns high confidence due to stronger local context. Such local confidence does not necessarily imply that the resulting global decoding trajectory is optimal. By exploring alternative position choices, SOAR can select less left-to-right-biased decoding paths when they receive higher confidence. We treat this as an empirical correlation rather than a causal explanation, and leave a more controlled study of the relationship between AR-ness, confidence, and decoding quality to future work.

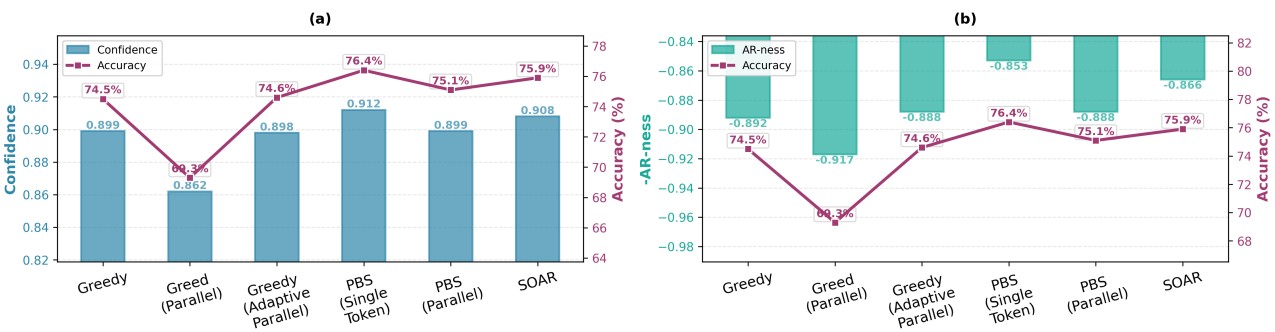

*Figure 6.* Left: Accuracy with average confidence. Right: Accuracy with negative Global AR-ness

*Table 2.* Results of other unmask metrics on HumanEval with DREAM-7B.

| Metric | Method | Length=256 | Length=512 |
|--------|--------|------------|------------|
| Margin | Greedy | 47.0 | 49.4 |
| Margin | SOAR | 51.2 | 51.2 |
| | *SpeedUp* | *×1.04* | *×1.07* |
| NegEntropy | Greedy | 51.2 | 56.1 |
| NegEntropy | SOAR | 53.0 | 57.9 |
| | *SpeedUp* | *×1.07* | *×1.14* |

### 4.5. Robustness to Unmask Metrics

Although softmax probability is the standard metric for selecting tokens to unmask, several alternative metrics are also viable options: (1) the margin between the top-1 and top-2 token probabilities and (2) the negative entropy of the probability distribution. Formally, given the token probability distribution $\mathbf{p} \in \mathbb{R}^V$, the two metrics are defined as

$$\text{Margin}(\mathbf{p}) = p_{(1)} - p_{(2)},$$

$$\text{NegEntropy}(\mathbf{p}) = \sum_{i=1}^{V} p_i \log p_i,$$

where $p_{(1)}$ and $p_{(2)}$ denote the largest and second-largest probabilities in $\mathbf{p}$, respectively.

Table 2 presents the results on HumanEval under two sequence length settings. We use $\tau = 0.9$ for margin-based metric and $\tau = -0.1$ for negative-entropy-based metric. For both, SOAR consistently improves accuracy over the corresponding greedy decoding baseline while maintaining a faster decoding speed.

### 4.6. Results on Instruction-Tuned Models

We further evaluate whether SOAR generalizes to instruction-tuned models, which are more practically relevant and may exhibit different confidence distributions.

Table 3 compares the token-level confidence distributions of LLADA-8B-Base and LLADA-8B-Instruct under greedy decoding on GSM8K.

*Table 3.* Token confidence distributions of LLADA-8B-Base and LLADA-8B-Instruct under greedy decoding on GSM8K. Differences are absolute percentage-point changes.

| Model | $[0, 0.8)$ | $[0.8, 0.995)$ | $[0.995, 1.0]$ |
|-------|-----------|----------------|----------------|
| LLADA-8B-Base | 34.4% | 17.4% | 48.2% |
| LLADA-8B-Instruct | 10.9% | 28.7% | 60.4% |
| *Difference* | *−23.5%* | *+11.3%* | *+12.2%* |

The instruction-tuned model is substantially more confident than the base model. In particular, the fraction of tokens with confidence below $0.8$ decreases by $23.5\%$, while the fraction of near-certain tokens with confidence at least $0.995$ increases by $12.2\%$. This shift raises a practical question: if most tokens already fall into a high-confidence regime, does SOAR still activate position search often enough to improve decoding?

Table 4 reports results on LLADA-8B-Instruct under both Pure Diffusion and Block Diffusion settings. We use the same confidence threshold as for LLADA-8B-Base, without retuning it for the instruction-tuned model. Despite the shifted confidence distribution, SOAR improves the average accuracy under both decoding settings while maintaining substantial speedups. Under Pure Diffusion, SOAR improves the average score from $49.1$ to $50.5$, with an average speedup of $×1.85$. Under Block Diffusion, SOAR improves the average score from $52.1$ to $53.6$, with an average speedup of $×1.89$.

The gains are not uniform across all individual settings. For example, SOAR slightly underperforms greedy decoding on HumanEval with maximum length $512$ under Block Diffusion. Nevertheless, the overall trend suggests that low-confidence positions remain useful decision points even for a more confident instruction-tuned model. These results

*Table 4.* Results on LLADA-8B-Instruct under Pure Diffusion and Block Diffusion (block size = 32) settings. SpeedUp is measured relative to standard greedy decoding within each setting.

| Setting | Method | HumanEval | | MBPP | | GSM8K | | Avg. |
|---|---|---|---|---|---|---|---|---|
| | Max Length | 256 | 512 | 256 | 512 | 256 | 512 | |
| Pure Diffusion | **Greedy** | 37.2 | 45.7 | 39.0 | 40.0 | 63.5 | 68.9 | 49.1 |
| | **SOAR** | 40.3(+3.1) | 45.7(+0.0) | 39.8(+0.8) | 41.4(+1.4) | 65.4(+1.9) | 70.3(+1.4) | 50.5(+1.4) |
| | *SpeedUp* | *×2.73* | *×2.09* | *×1.31* | *×1.08* | *×1.80* | *×2.11* | *×1.85* |
| Block Diffusion | **Greedy** | 37.8 | 48.8 | 34.4 | 36.6 | 77.6 | 77.5 | 52.1 |
| | **SOAR** | 39.6(+1.8) | 46.9(-1.9) | 39.2(+4.8) | 39.4(+2.8) | 78.1(+0.5) | 78.5(+1.0) | 53.6(+1.5) |
| | *SpeedUp* | *×2.47* | *×2.07* | *×1.04* | *×1.09* | *×2.04* | *×2.63* | *×1.89* |

indicate that confidence-switched position search can generalize beyond base models, while also highlighting that confidence thresholds are model-dependent rather than universally optimal.

We further provide a compute-matched comparison with best-of-$N$ sampling in Appendix A.3, showing that step-wise position search is a more efficient use of additional compute than independent resampling.

## Limitations

Although SOAR improves average generation quality in our experiments, several limitations remain.

**Local confidence is not sequence-level correctness.** SOAR ranks candidate paths using average token-level confidence. This criterion can favor an incorrect branch when that branch contains shorter or more locally confident token decisions than the correct one. In such cases, token-level confidence acts as an imperfect proxy for global sequence quality and can be misled by short-term confidence spikes. Representative failure cases are provided in Appendix C. A promising direction is to compare candidate branches using span-level confidence, verifier scores, or other sequence-level estimates that better capture the long-term effect of an unmasking decision.

**Search may be under-activated on overconfident models.** SOAR relies on low-confidence positions to trigger position beam search. When token confidences are highly saturated near 1.0, SOAR may degenerate toward parallel decoding and lose part of the benefit of position search. This issue may arise for RL-optimized DLMs (Ni et al., 2026), where adaptive threshold selection or confidence recalibration could help.

**Model-dependent confidence threshold.** Although we use a single threshold for each model family rather than tuning a separate threshold for each benchmark, the threshold still differs across model families: $\tau = 0.95$ for LLADA-8B

and $\tau = 0.90$ for DREAM-7B. This reflects differences in their confidence distributions. Applying SOAR to new DLM families may therefore require a lightweight calibration step.

## Conclusion

We introduce **S**earch **O**r **A**ccele**R**ate (SOAR), a training-free decoding framework for Diffusion Language Models that adaptively balances position-space search and parallel decoding. When the model is uncertain, SOAR widens the search over alternative unmasking decisions; when the model is confident, it collapses the search and decodes high-confidence positions in parallel. This confidence-switched strategy improves the quality–speed trade-off of DLM inference across mathematical reasoning and code generation benchmarks. Our experiments show that position search improves generation quality but can be computationally expensive, while SOAR preserves much of this benefit with substantially lower inference cost. Additional evaluations on instruction-tuned models and compute-matched best-of-$N$ baselines further support the practical value of allocating search only to uncertain decoding steps. More broadly, SOAR suggests that token confidence is not only a diagnostic signal in DLM decoding, but also an actionable signal for controlling when to search and when to accelerate.

## Acknowledgements

This work was supported by the AI Research Resource (AIRR) under grant number 0261-6962-8988-1.

## Impact Statement

This paper presents work whose goal is to advance the field of Machine Learning. There are many potential societal consequences of our work, but we do not identify any that require specific mitigation beyond standard considerations for language models.

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

# A. Additional Experiments

## A.1. SOAR as a Universal Decoding Framework

SOAR is a training-free, search-based inference framework that operates on the choice of unmasking positions without modifying model parameters or model architecture. This makes it largely orthogonal to decoding methods that control early termination, variable generation length, or caching behavior. We evaluate this complementarity with two representative families below.

**Compatibility with early-termination methods.** Prophet (Li et al., 2025b) reduces decoding cost by terminating trajectories early when the model already predicts the final answer. Table 5 reports results on LLADA-8B-Instruct on HumanEval and GSM8K. Combining SOAR with Prophet improves over Prophet alone on both benchmarks while preserving substantial speedup over greedy decoding. On GSM8K, SOAR+Prophet does not exceed the original greedy accuracy, but it substantially recovers the quality drop introduced by Prophet while still accelerating inference.

*Table 5.* Results of combining SOAR with Prophet on GSM8K and HumanEval.

| Method | HumanEval | GSM8K |
|---|---|---|
| Greedy | 37.8 | **77.5** |
| Prophet | 37.5 | 75.4 |
| *SpeedUp* | *×1.80* | *×1.20* |
| SOAR + Prophet | **40.2** | 77.2 |
| *SpeedUp* | *×2.27* | *×1.69* |

**Compatibility with variable-length decoding.** To assess the robustness of SOAR when combined with variable-length decoding, we evaluate it on DREAMON (Wu et al., 2026), a model post-trained upon DREAM-7B that dynamically adjusts sequence length during decoding by predicting *expand* and *delete* tokens. When an *expand* token is generated, it is replaced with two mask tokens, thereby extending the sequence; when a *delete* token is generated, the corresponding token is removed from the output sequence.

Following the DREAMON setup, we evaluate on the HumanEval-Infilling benchmark (Bavarian et al., 2022). We set the maximum decoding length to 64, use negative entropy with $\tau = -0.1$ as the confidence metric, and report exact match accuracy. Table 6 compares greedy decoding and SOAR across different initial mask lengths. SOAR consistently improves exact match accuracy while maintaining nearly the same inference speed. This suggests that position search can remain effective even when the output length is not fixed, further supporting the compatibility of SOAR with other DLM decoding mechanisms.

*Table 6.* Results of combining SOAR with DREAMON on HumanEval-Infilling task.

| Method | Initial Mask Length | | | |
|---|---|---|---|---|
| | 8 | 16 | 32 | 64 |
| Greedy | 58.4 | 57.4 | 58.0 | 58.0 |
| SOAR | **63.0** | **66.1** | **67.2** | **67.7** |
| *SpeedUp* | *×1.02* | *×1.02* | *×1.03* | *×1.04* |

## A.2. Comparison with Other DLM Decoding Methods

**Comparison with search-based decoding.** The most closely related concurrent work is Order-Token Search (Shen et al., 2026), which explicitly searches over both the generation order and the token space. This broader search space can provide stronger quality improvements, since the method can revise not only *which positions* to unmask but also *which tokens* to consider during decoding. However, this additional flexibility also increases computational cost: searching jointly over the order and token dimensions requires maintaining and scoring a larger set of candidates, leading to slower inference.

A fully controlled empirical comparison is difficult because the implementation of Order-Token Search is not publicly released. We therefore avoid presenting a direct leaderboard-style table based on potentially mismatched implementations.

Instead, we position the two methods by their design goals. Order-Token Search focuses on improving generation quality through a larger search space, even when this incurs substantial decoding overhead. In contrast, SOAR searches only over the position dimension and activates this search only when the model is uncertain. When confidence is high, SOAR collapses the beam and decodes multiple positions in parallel. This design targets a different operating point: *improving generation quality without sacrificing inference speed*.

In this sense, the two methods are complementary examples of search-based DLM decoding. Order-Token Search explores a more expressive but more expensive search space, while SOAR focuses on a confidence-switched search rule that preserves the practical speed advantage of parallel DLM decoding.

**Comparison with acceleration methods.** SlowFast Sampling (Wei et al., 2026) and WINO (Hong et al., 2025) are designed primarily for accelerating DLM inference. They improve efficiency by changing the denoising schedule, reducing the number of effective decoding steps, or managing token commitments more aggressively. This objective differs from that of SOAR, which uses confidence-guided position search to improve generation quality while preserving competitive decoding speed.

Although these acceleration methods are conceptually complementary to position-level search, directly combining them with SOAR is non-trivial in practice. Their acceleration behavior is tightly coupled with their own decoding logic and hyperparameter choices, whereas SOAR decides when to search or parallel-decode based on token confidence. We therefore treat SlowFast Sampling and WINO as comparison points rather than combination partners, and use them to clarify where SOAR lies on the quality–speed Pareto frontier.

For comparison, we evaluate all methods under the same model and decoding configuration: LLaDA-8B-Instruct with Block Diffusion, block size 32, maximum length 256, on GSM8K. Since WINO uses its own evaluation pipeline, we report results under both the official WINO codebase and lm-evaluation-harness. The two columns should be interpreted only as within-framework comparisons, as prompt templates and post-processing differ across frameworks.

As shown in Table 7, SlowFast Sampling and WINO generally achieve larger speedups than SOAR, reflecting their focus on throughput. However, under lm-evaluation-harness, these speedups are accompanied by accuracy drops relative to greedy decoding. In contrast, SOAR is not optimized for maximal acceleration, but it consistently improves accuracy over greedy decoding in both evaluation settings while still providing a meaningful speedup. This suggests that SOAR occupies a different region of the quality–speed Pareto frontier: acceleration-focused methods prioritize throughput with secondary attention to quality preservation, whereas SOAR prioritizes quality improvement through confidence-guided search while retaining competitive decoding speed.

*Table 7.* Comparison with SlowFast Sampling and WINO on GSM8K. Results are reported under both the official WINO codebase and lm-evaluation-harness.

| Method | WINO codebase | lm-eval-harness |
| --- | --- | --- |
| Greedy | 78.9 | 77.6 |
| SlowFast Sampling | — | 75.1(-2.5) |
| *SpeedUp* | *—* | *×2.93* |
| WINO | 81.2(+2.3) | 76.7(-0.9) |
| *SpeedUp* | *×2.51* | *×3.67* |
| SOAR (Ours) | **82.3(+3.4)** | **78.1(+0.5)** |
| *SpeedUp* | *×1.41* | *×2.04* |

### A.3. Compute-Matched Comparison with Best-of-$N$

A natural question is whether the gains of position search simply come from spending additional compute. We therefore compare against a compute-matched best-of-$N$ baseline, which generates $N$ independent sequences and selects the one with the highest average confidence. With $N = 2$, best-of-$N$ uses a similar two-trajectory budget to PBS with beam size $K = 2$, but allocates this budget to independent resampling rather than step-wise position decisions.

Table 8 reports Pass@1, inference time, and average confidence on HumanEval using LLaDA-8B-Instruct with Block Diffusion, block size 32, and maximum length 256. Results are averaged over three runs. To obtain diverse samples, best-of-$N$ uses temperature 0.5; all other methods use temperature 0. Under the matched computation, PBS achieves a

higher Pass@1 point estimate than best-of-$N$ and also yields higher average confidence. This suggests that allocating compute to step-wise position search is more effective than generating independent samples and selecting one post hoc. However, PBS roughly doubles inference time relative to greedy decoding.

By contrast, SOAR activates search only at low-confidence steps and collapses to parallel decoding when the model is confident. As a result, it achieves accuracy comparable to PBS while reducing inference time from 45.6 minutes to 9.1 minutes. It is also faster than greedy decoding in this setting. These results support the central design of SOAR: additional compute is most useful when concentrated on uncertain position decisions rather than spent uniformly across the whole decoding process.

*Table 8.* Comparison with best-of-$N$ sampling on HumanEval (LLADA-8B-Instruct, Block Diffusion, block size $= 32$, max length $= 256$). Results are averaged over 3 runs.

|  | **Greedy** | **Best-of-$N$** | **PBS** | **SOAR** |
|---|---|---|---|---|
| Pass@1 | $37.4{\pm}2.3$ | $37.8{\pm}3.4$ | $\mathbf{38.7}{\pm}0.5$ | $38.6{\pm}2.4$ |
| Inference Time (min) | $23.2{\pm}2.6$ | $46.3{\pm}2.6$ | $45.6{\pm}1.3$ | $\mathbf{9.1}{\pm}0.1$ |
| Average Confidence | $0.956{\pm}.000$ | $0.961{\pm}.001$ | $\mathbf{0.965}{\pm}.001$ | $0.963{\pm}.001$ |

# B. Visualization of SOAR

To better understand the dynamic behavior of `SOAR` during inference, we provide two visualizations. The first analyzes how the two decoding modes evolve across the decoding process at a population level; the second illustrates a concrete decoding trajectory on a single example, contrasting `SOAR` with standard greedy decoding.

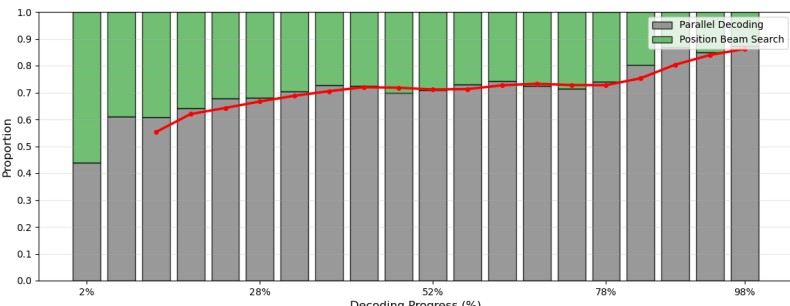

*Figure 7.* Variation of Decoding Modes Across the Decoding Process. The x-axis represents the progress of decoding (quantized into 20 bins), while the y-axis indicates the proportion of tokens decoded by each mode across all samples. The stacked bars are partitioned into gray (Parallel Decoding) and green (PBS) segments, reflecting their relative usage at each stage. A red trend line highlights the overall transition pattern between the two modes.

**Increasingly Parallel Decoding.** We analyzed the usage patterns of Beam Search and Parallel Decoding modes across decoding steps on the DREAM-7B-Base model using GSM8K. Consistent with the processing in Subsection 4.4, for all sample decoding sequences, we retain only the valid portion — i.e., the tokens preceding the keyword "answer". Since different samples have varying effective decoding lengths $S_{\text{sample}}$, we record the ratio of each token's decoding step to its effective sequence length as a measure of the Decoding Process(%). Figure 7 illustrates how the two decoding modes evolve as decoding progresses. In the early stages, SOAR allocates a larger proportion of steps (55%) to PBS to explore more promising sequences. As decoding advances, the model eventually favors parallel decoding in 85% of the cases. This trend demonstrates the model's transition from initial uncertainty toward greater confidence. Throughout the inference process, SOAR dynamically switches between the two decoding modes based on confidence scores in an adaptive manner.

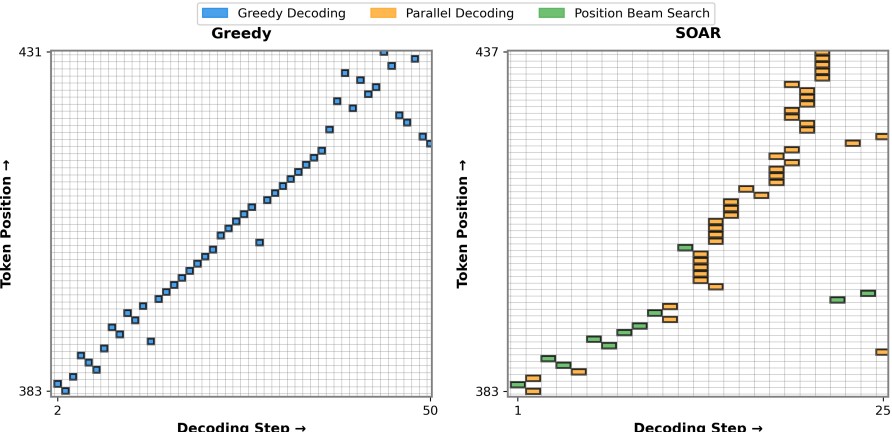

*Figure 8.* Decoding Path Comparison. The left figure shows Greedy Decoding, and the right figure shows Our method. The X-axis represents steps, and the Y-axis represents positions.

**Decoding Path Illustration**. Figure 8 illustrates the decoding trajectories of a GSM8K sample using Greedy and SOAR decoding, based on the DREAM-7B-Base model. The Greedy decoder exhibits an AR-style decoding path, while SOAR, through parallel decoding, decodes nearly the same number of positions in fewer steps. Moreover, by employing PBS, SOAR retains earlier positions as masked and defers their decoding to later steps, aligning with the analysis in Subsection 4.4. For tokens that appear early but lack sufficient model confidence, decoding them later with richer contextual information yields an overall higher-confidence sequence compared to Greedy decoding.

# C. Failure Case Analysis

To better understand when SOAR can underperform greedy decoding, we manually inspect representative code-generation examples where greedy decoding produces a correct program but SOAR fails. Table 9 summarizes three typical failure patterns. In all cases, the failure is not caused by a single isolated token error. Instead, SOAR selects a locally high-confidence branch whose program structure is globally incorrect. Once this branch is committed, subsequent denoising steps tend to complete a coherent but wrong program, making later recovery difficult.

Overall, two cases are mainly associated with early wrong position choices that commit to an incorrect control-flow structure, while one case reflects a later-stage premature closure of the function. This supports the limitation discussed in the main text: token-level confidence is useful for guiding search, but it is not always a reliable proxy for sequence-level program correctness.

*Table 9.* Representative failure cases where SOAR underperforms greedy decoding. We show key generated branches rather than full programs. The common pattern is that SOAR selects a locally high-confidence branch that is globally incorrect. Red text marks the branch most directly associated with the failure.

| Prompt & issue | Greedy output | SOAR output |
|---|---|---|
| **Largest divisor.** Find the largest number that divides $n$ evenly and is smaller than $n$. 

 *Issue.* Early wrong branch. SOAR commits to a high-confidence even-number shortcut, which is not the largest proper divisor for most even $n$. | ```if n <= 1:\n  return 1\nfor i in range(n - 1, 0, -1):\n  if n % i == 0:\n    return i\nreturn 1``` | ```if n <= 1:\n  return 1\nif n % 2 == 0:\n  return 2\ni = 3\nwhile i * i <= n:\n  if n % i == 0:\n    return i\n  i += 2\nreturn n``` |
| **Starts or ends with 1.** Count the $n$-digit positive integers that start or end with 1. 

 *Issue.* Early structural error. SOAR chooses an incorrect loop range and indexes an integer as if it were a string, so later denoising cannot recover the missing enumeration logic. | ```count = 0\nfor i in range(10**(n-1), 10**n):\n  if (str(i).startswith('1') or\n    str(i).endswith('1')):\n    count += 1\nreturn count``` | ```count = 0\nfor i in range(10**n, 10**n + 1):\n  if i[-1] == '1' or i[0] == '1':\n    count += 1\nreturn count``` |
| **Conditional sum.** Square indices divisible by 3, cube indices divisible by 4 but not 3, and keep all other elements unchanged. 

 *Issue.* Later premature closure. SOAR captures the two special cases but omits the residual else branch, then closes the function too early. | ```total = 0\nfor i in range(len(lst)):\n  if i % 3 == 0:\n    total += lst[i] ** 2\n  elif i % 4 == 0:\n    total += lst[i] ** 3\n  else:\n    total += lst[i]\nreturn total``` | ```total = 0\nfor i in range(len(lst)):\n  if i % 3 == 0:\n    total += lst[i] ** 2\n  elif i % 4 == 0 and i % 3 != 0:\n    total += lst[i] ** 3\nreturn total``` |

These examples suggest that the main failure mode of SOAR is not simply insufficient exploration, but imperfect branch scoring. Average token-level confidence can prefer short, familiar, or syntactically natural code fragments even when they encode the wrong algorithm. This motivates future work on span-level or verifier-based candidate scoring, where candidate branches are compared according to their longer-range semantic effect rather than only their local token confidence.

# D. Pseudocode of SOAR

Algorithm 1 presents the complete decoding procedure of `SOAR`, which dynamically switches between parallel decoding and position beam search based on token-level confidence.

---

**Algorithm 1 S**earch **O**r **AcceleR**ate (`SOAR`)

---

**Require:** Prompt $x_{\text{prompt}}$, DLM $f_\theta$, max length $L$, confidence threshold $\tau$, max beam width $B_{\text{max}}$
**Ensure:** Generated sequence $x_0^\star$
 1: Initialize $x_T$ with $x_{\text{prompt}}$ followed by $L$ [MASK] tokens
 2: Initialize beam $\mathcal{B}_T \leftarrow \{(x_T, 0, \text{INIT})\}$
 3: **for** $t = T, T-1, \ldots, 1$ **do**
 4:   $\mathcal{C}_t \leftarrow \emptyset$ {Candidate set for the next beam}
 5:   **for** each candidate $(x_t, s_t, \cdot) \in \mathcal{B}_t$ **do**
 6:    $\mathbf{p}_t \leftarrow f_\theta(x_t)$
 7:    $M_t \leftarrow \{i : x_{t,i} = [\text{MASK}]\}, \quad c_{t,i} \leftarrow \max_v \mathbf{p}_{t,i}[v]$
 8:    $H_t \leftarrow \{i \in M_t : c_{t,i} > \tau\}$
 9:    **if** $H_t \neq \emptyset$ **then**
10:     Add PARALLELEXPAND$(x_t, s_t, H_t, \mathbf{p}_t)$ to $\mathcal{C}_t$ {unmask all high-confidence positions}
11:    **else**
12:     Add PBSEXPAND$(x_t, s_t, M_t, \mathbf{p}_t, B_{\text{max}})$ to $\mathcal{C}_t$ {search over top uncertain positions}
13:    **end if**
14:   **end for**
15:   Sort $\mathcal{C}_t$ by decoding score in descending order
16:   Let $(x^\star, s^\star, m^\star)$ be the top candidate in $\mathcal{C}_t$
17:   **if** $m^\star = \text{PARALLEL}$ **then**
18:    $N \leftarrow 1$ {high confidence: collapse to one candidate}
19:   **else**
20:    $N \leftarrow \min(B_{\text{max}}, |\mathcal{C}_t|)$ {uncertainty: keep multiple candidates}
21:   **end if**
22:   $\mathcal{B}_{t-1} \leftarrow$ top $N$ candidates from $\mathcal{C}_t$
23: **end for**
24: $(x_0^\star, s_0^\star, \cdot) \leftarrow \arg\max_{(x,s,m) \in \mathcal{B}_0} s \; x_0^\star = 0$

---

