# OpenReview forum: "Search or Accelerate: Confidence-Switched Position Beam Search for Diffusion Language Models"
_ICML.cc/2026/Conference — ICML 2026 regular_

### Official Review · Reviewer_T5Gd · 2026-02-25

**Soundness:** 3
**Presentation:** 3
**Significance:** 2
**Originality:** 2
**Overall Recommendation:** 4
**Confidence:** 3

**Summary:**

A fundamental question considered by this study is whether diffusion language model decoding can be improved by searching over unmasking orders rather than always greedily unmasking the most confident positions. The paper proposes a training-free framework, SOAR, that adaptively switches between position beam search when confidence is low and parallel decoding when confidence is high, aiming to balance output quality and inference speed. Overall, the research's fundamental contribution is introducing this confidence-switched decoding strategy, along with the underlying Position Beam Search idea, and showing it improves reasoning/code generation performance across DLM benchmarks while remaining efficient in practice.

**Compliance With Llm Reviewing Policy:**

Affirmed.

**Final Justification:**

Thank you for the author's detailed reply, which has resolved my issue. Since my score is already at the acceptance threshold and I don't have a very deep understanding of this research area, I'm keeping my score and confidence.

**Key Questions For Authors:**

1. Can you provide a breakdown of cases where SOAR underperforms greedy/adaptive parallel decodin, and whether these failures are associated with early wrong position choices or later-stage recovery issues?
2. How sensitive is SOAR to threshold/beam settings when moving across tasks within the same model, and were the reported settings selected globally or per-benchmark?

**Limitations:**

yes

**Strengths And Weaknesses:**

Strengths
1. The paper is methodologically careful in separating the contribution into two layers (PBS and SOAR), which makes it easier to verify what each component contributes to the final quality–speed tradeoff.
2. The paper has a strong narrative arc: it starts from a concrete limitation of confidence-based parallel decoding, introduces position-space search as a targeted remedy, and then motivates SOAR as the practical compromise.

Weaknesses
1. Compared with confidence-based adaptive-parallel methods, which SOAR explicitly uses as a foundation, the paper still relies on raw max-softmax confidence as the central switching signal, but does not provide a calibration-oriented analysis, e.g., when confidence is spuriously high yet the decoding path is wrong.
2. The analysis in Section 4.4 provides useful correlations: confidence / AR-ness vs. accuracy, but the “entropy sink” explanation is presented as a hypothesis rather than being tested with stronger causal controls.
3. The dynamic candidate-size rule is a hard reset to 1 or K based on the origin of the best new candidate, which is a reasonable heuristic, but the paper does not deeply justify why this binary rule is preferable to softer or score-gap-based alternatives.

---

> ### Author Rebuttal · Authors · 2026-03-31
>
> **Response to Reviewer T5Gd**
>
> We sincerely thank the reviewer for the detailed and constructive feedback. We address the comments below.
>
> ---
>
> >**W1. Confidence Calibration and Entropy Sink Analysis**
>
> We acknowledge that the “entropy sink” explanation in Section 4.4 is a hypothesis, not a causal claim. Our understanding is informed by DiffuCoder [1], which observes that DLMs bias toward tokens right of the prefix. We hypothesize that lower Global AR‑ness may mitigate this issue. We will clarify this in the camera‑ready version and outline future validation.
>
> ---
>
> >**W2. Dynamic Candidate-Size Rule**
>
> We appreciate the suggestion. The binary reset rule is a heuristic based on the observation that confidence increases during decoding (Figure 7). We explored a step‑based schedule (candidate‑size 3/2/1 for steps <50/50-100/>100), but found no significant accuracy gain (Table 4‑1). Since decoding steps vary across samples under adaptive parallel decoding setting, a proportion‑based schedule is infeasible. We agree that score-gap-based adjustment is a promising direction and will explore it in future work.
>
> Table 4-1: Results of step-based soft candidate size schedule
>
> | Method | HumanEval | MBPP | GSM8K |
> |--------|:-:|:-:|:-:|
> | Fixed candidate size | 55.5 | 57.0 | 74.7 |
> | Step-based soft candidate size | 54.7 | 56.5 | 74.9 |
>
>
> ---
>
> >**Q1. Failure Case Analysis**
>
>  We thank the reviewer for this insightful question. In Table 4‑2, we provide several cases where SOAR underperforms compared to greedy decoding. For each case, we recorded the full generation trajectory and confidence values (omitted from the table due to length) and analyzed the root cause of the failure.
>
> Across the analyzed cases, the failure pattern is consistent: **incorrect branches are favored because they are shorter or contain more locally confident tokens**, leading to higher average confidence at the decision point.
>
> We note that SOAR optimizes for local average confidence at each step rather than global sequence confidence. This is a limitation of token‑level confidence‑based selection: a shorter or locally confident incorrect branch can be preferred over a longer correct branch that requires more steps to reach high confidence.
>
> These observations motivate a potential improvement: **moving from token‑level confidence to span‑level confidence** when comparing candidate branches. By evaluating the confidence of a contiguous span of tokens, the model could better estimate the long‑term quality of a decision rather than being misled by short‑term confidence spikes.
>
> We believe this insight is valuable for future work on search‑based decoding for diffusion language models and will include a discussion in the camera‑ready version.
>
> Table 4-2: Cases Where SOAR Underperforms Greedy Decoding on HumanEval
>
> | Problem | Greedy | SOAR | Issue |
> |---------|---------------|-------------|-------|
> | Largest divisor | Correct implementation with descending loop | Incorrect: `if n % 2 == 0: return 2` | The incorrect branch has higher confidence |
> | Starts one ends | Correct: `str(i).startswith('1') or str(i).endswith('1')` | Incorrect: `i[-1] == '1' or i[0] == '1'` | The incorrect branch has higher confidence |
> | Sum squares | Correct with `else: total += lst[i]` | Missing `else` branch | `return total` has higher confidence than `else` |
>
> ---
>
> >**Q2. Hyperparameter Sensitivity**
>
> We thank the reviewer for the question. We selected hyperparameters **globally** across tasks for the same model, aiming for a configuration that works robustly across benchmarks.
>
> - Beam size: We observed consistent trends across multiple datasets. While increasing beam size beyond 2 yields only marginal accuracy gains, it significantly slows down inference (as shown in Figure 5). Therefore, we set `max_beam_size=2` globally. Beam size is not a sensitive hyperparameter, though excessively large values clearly hurt decoding speed.
>
> - Threshold: We acknowledge that the confidence threshold may be more sensitive, as confidence distributions can vary across tasks. However, our ablation study in Figures 4(b) and 4(c) of the paper shows that thresholds capable of improving quality are largely similar across MBPP and GSM8K (≥0.8). We evaluated candidate thresholds (0.8, 0.9, 0.95, 0.98) and selected the one with the highest average accuracy across tasks. As shown in Figure 4(c), although τ=0.95 performs better on GSM8K, we reported τ=0.9 in the main results as it achieves the best overall trade‑off across benchmarks.
>
> We believe this global selection demonstrates SOAR's robustness across different tasks without requiring per‑benchmark tuning.
>
> ---
>
> >**References**
>
> [1] Gong, Shansan, et al. "DiffuCoder: Understanding and Improving Masked Diffusion Models for Code Generation." *arXiv preprint arXiv:2506.20639* (2025).

---

> > ### Author Rebuttal · Reviewer_T5Gd · 2026-04-05
> >
> > Thank you for the author's detailed reply, which has resolved my issue. Since my score is already at the acceptance threshold and I don't have a very deep understanding of this research area, I'm keeping my score and confidence.

---

> > > ### Author Response · Authors · 2026-04-06
> > >
> > > Thank you very much for the follow-up and for your careful consideration of our rebuttal. We are glad that our response helped resolve your concerns, and we sincerely appreciate your thoughtful assessment and feedback. Thank you again for your time and consideration.
> > >
> > > Sincerely,
> > >
> > > The Authors

---

### Official Review · Reviewer_HW4Z · 2026-03-05

**Soundness:** 2
**Presentation:** 2
**Significance:** 2
**Originality:** 2
**Overall Recommendation:** 4
**Confidence:** 4

**Summary:**

The paper proposes Search Or AcceleRate (SOAR), a training-free decoding algorithm designed for Diffusion Language Models (DLMs). The core mechanism dynamically alternates between parallel decoding and position beam search (PBS) based on a token-level confidence threshold. When the model exhibits high confidence, it decodes multiple positions in parallel to accelerate inference ; when confidence falls below a threshold τ, it reverts to a beam search in the position space to avoid suboptimal unmasking orders.

**Compliance With Llm Reviewing Policy:**

Affirmed.

**Final Justification:**

Basically satisfied with the authors’ response. I appreciate the additional experiments and clarifications, and I am inclined to increase my score.

**Key Questions For Authors:**

Why are the results of LLaDA on GSM8K and HumanEval much lower than those reported in other papers?

**Strengths And Weaknesses:**

Strength

- The transition from vocabulary-space beam search (standard in AR models) to position-space search is a logical and creative adaptation of classical search algorithms to non-autoregressive paradigms.
- The dynamic switching mechanism based on a simple confidence threshold is intuitive and easily pluggable into existing mask-based DLMs without requiring architectural modifications or tuning.

Weaknesses

- As shown in Table 1, this paper fails to establish a dominant speed-quality Pareto frontier.
- The paper evaluates solely against basic Greedy and PBS baselines. It completely ignores state-of-the-art DLM inference methods.
- The method seems to be sensitive to the hyper parameters.

---

> ### Author Rebuttal · Authors · 2026-03-31
>
> **Response to Reviewer HW4Z**
>
> We sincerely thank the reviewer for the detailed and constructive feedback. We address the comments below.
>
> >**W1. Pareto Frontier**
>
> Thank you for this important point. We agree that a clear Pareto frontier is essential. As shown in Figure 3, while PBS achieves higher quality, it incurs significant slowdown. SOAR improves quality over the baseline while maintaining competitive speed, and we believe it represents a favorable point on the quality-speed Pareto frontier.
>
> >**W2. Lack of comparison with other methods**
>
> We thank the reviewer for the suggestion. We would like to clarify that SOAR is designed as a training‑free, search‑based inference framework that can be **combined with other state‑of‑the‑art** methods to further improve generation quality and efficiency.
>
> - Acceleration methods (e.g., Prophet [1], Table 3‑1) reduce decoding steps by terminating trajectories early. SOAR could further improves efficiency and quality. We note that the speedup ratios in this table may appear slightly inconsistent with those in other tables (e.g., Table 2‑2) in the rebuttal. This is because, to faithfully reproduce the Prophet results, all experiments in this table were conducted using the `lm-eval harness` framework for inference, whereas others used `OpenCompass`. The two frameworks have minor differences in their implementation. However, we ensure that within each table, all evaluations are performed under **the same framework**.
>
> - Variable‑length decoding methods (e.g., DreamOn [2], Table 3-2) adjust sequence length on the fly. SOAR remains effective when combined with DreamOn across multiple initial mask lengths.
>
> Table 3-1: Results on combining SOAR with Prophet on LLaDA-8B-Instruct
>
> | Method | HumanEval | GSM8K |
> |--------|:-:|:-:|
> | LLaDA-Instruct-8B | 37.8 | 77.5 |
> | Prophet | 37.5 | 75.4 |
> | SpeedUp | ×1.80 | ×1.20 |
> | SOAR+Prophet | 40.2 | 77.2 |
> | SpeedUp | ×2.27 | ×1.69 |
>
> Table 3-2: Results on combining SOAR with DreamOn on DreamOn-v0-7B
>
> | Method | 8 | 16 | 32 | 64 |
> |--------|:-:|:-:|:-:|:-:|
> | Greedy | 58.4 | 57.4 | 58.0 | 58.0 |
> | SOAR | 63.0 | 66.1 | 67.2 | 67.7 |
> | SpeedUp | ×1.02 | ×1.02 | ×1.03 | ×1.04 |
>
>
> SOAR’s true strength lies in its complementarity with other DLM inference advances. We will add a discussion in the camera‑ready version to guide future work on combining SOAR with other methods.
>
>
> >**W3. Hyperparameter Sensitivity**
>
>
> - **Beam size.** SOAR is not sensitive to beam size. As shown in Figure 5, increasing beam size beyond 2 yields only marginal accuracy gains while significantly slowing down inference. We therefore set `beam_size=2` as the default across all experiments.
>
> - **Confidence threshold.** As shown in the ablation study (Figure 4), a wide range of thresholds (0.8–0.95) consistently achieve quality improvement while maintaining or exceeding greedy decoding speed. This demonstrates that SOAR does not require careful per-task threshold tuning to be effective.
>
>
> >**Q1. Results with Block Diffusion**
>
> We would like to clarify that the state-of-art works (e.g. Prophet[1], Fast-dllm[3]) use **Block Diffusion** inference (semi-autoregressive) to achieve better performance on GSM8K and HumanEval. However, to maintain consistency with the **official LLaDA experimental setting** [4] (Page 7, Table 1), our experiments mainly use **Pure Diffusion**.
>
> To further demonstrate the robustness of our method, we conducted supplementary experiments on LLaDA-8B-Instruct with Block Diffusion settings, implemented based on the OpenCompass to ensure consistency. As shown in Table 3-3, the quality improvement without sacrificing speed generalizes well to the block diffusion setting.
>
> Table 3-3: Results on LLaDA-8B-Instruct using Block Diffusion (Block size=32)
>
> | | HumanEval (256) | HumanEval (512) | MBPP (256) | MBPP (512) | GSM8K (256) | GSM8K (512) |
> |---|:-:|:-:|:-:|:-:|:-:|:-:|
> | Greedy | 37.8 | 48.8 | 34.4 | 36.6 | 77.6 | 77.5 |
> | SOAR | 39.6 (+1.8) | 46.9 (−1.9) | 39.2 (+4.8) | 39.4 (+3.8) | 78.1 (+0.5) | 78.5 (+1.0) |
> | SpeedUp | ×2.47 | ×2.07 | ×1.04 | ×1.09 | ×2.04 | ×2.63 |
>
>
>
> >**References**
>
> [1] Li, Pengxiang, et al. "Diffusion language models know the answer before decoding." *arXiv preprint arXiv:2508.19982* (2025).
>
> [2] Wu, Zirui, et al. "DreamOn: Diffusion Language Models For Code Infilling Beyond Fixed-size Canvas." *arXiv preprint arXiv:2602.01326* (2026).
>
> [3] Wu, Chengyue, et al. "Fast-dllm: Training-free acceleration of diffusion llm by enabling kv cache and parallel decoding." *arXiv preprint arXiv:2505.22618* (2025).
>
> [4] Nie, Shen, et al. "Large language diffusion models." arXiv preprint arXiv:2502.09992 (2025).

---

> > ### Author Rebuttal · Reviewer_HW4Z · 2026-04-02
> >
> > I remain highly concerned about the claimed superiority of the experimental results. Taking GSM8K as an example, [1] achieves a 3.2× speedup and [2] achieves a 6× speedup, whereas SOAR provides less than a 2× speedup in comparison.
> >
> > [1]https://arxiv.org/abs/2506.10848
> >
> > [2]https://arxiv.org/abs/2507.18578

---

> > > ### Author Response · Authors · 2026-04-06
> > >
> > > Thank you for pointing out these strong baselines. We have conducted additional experiments to provide a comprehensive comparison.
> > >
> > > **Positioning of SOAR.** We would like to clarify that SOAR and methods like SlowFast Sampling [1] and WINO [2] target different points on the quality–speed Pareto frontier. SlowFast and WINO are primarily designed to accelerate DLM inference, whereas SOAR is designed to *improve generation quality* through confidence-guided position search, while preserving competitive speed via adaptive parallel decoding. We view these as complementary rather than competing goals.
> > >
> > > **Direct comparison under a unified setting.** To ensure a fair comparison, we evaluated all methods under the same configuration (length=256, block=32, LLaDA-8B-Instruct). Since WINO uses its own evaluation pipeline, we report results under both the official WINO codebase and lm-eval-harness. The discrepancy between the two frameworks likely stems from differences in prompt templates and post-processing, which affects all methods consistently.
> > >
> > > **Table 3-3:** Comparison with SlowFast Sampling and WINO on GSM8K
> > >
> > > | Method | WINO-official-code | lm-eval-harness |
> > > |---|---|---|
> > > | LLaDA-8B-Instruct | 78.9 | 77.6 |
> > > | SlowFast Sampling | - | 75.1 (−2.5) |
> > > | *SpeedUp* | - | ×2.93 |
> > > | WINO | 81.2 (+2.3) | 76.7 (−0.9) |
> > > | *SpeedUp* | ×2.51 | ×3.67 |
> > > | **SOAR (Ours)** | **82.3 (+3.4)** | **78.1 (+0.5)** |
> > > | *SpeedUp* | ×1.41 | ×2.04 |
> > >
> > > The results show that different methods excel in different dimensions: SlowFast and WINO achieve significant speedups, while SOAR consistently achieves the highest accuracy under both evaluation settings. This highlights that these methods target different regions of the quality–speed Pareto frontier, and SOAR fills a complementary niche where generation quality is the primary objective.
> > >
> > > **Complementarity with acceleration methods.** We note that SOAR operates on a different axis from these acceleration techniques, it focuses on *which positions* to unmask, rather than *how fast* to unmask them. As demonstrated in Table 3-1 of our previous response, this orthogonality has already enabled successful combination with Prophet, yielding both higher quality and faster speed (e.g., on HumanEval: 40.2% at ×2.27 vs. Prophet's 37.5% at ×1.80). We believe combining SOAR with methods like WINO or SlowFast is a promising direction, and we will discuss these comparisons in the revised manuscript.
> > >
> > > [1] Wei, Qingyan, et al. "Accelerating diffusion large language models with slowfast sampling." arXiv preprint arXiv:2506.10848 (2025).
> > >
> > > [2] Hong, Feng, et al. "Wide-in, narrow-out: Revokable decoding for efficient and effective DLLMs." arXiv preprint arXiv:2507.18578 (2025).

---

### Official Review · Reviewer_hkNQ · 2026-03-15

**Soundness:** 3
**Presentation:** 3
**Significance:** 3
**Originality:** 3
**Overall Recommendation:** 4
**Confidence:** 3

**Summary:**

This paper proposes SOAR (Search Or AcceleRate), a training-free decoding algorithm for Diffusion Language Models (DLMs) that adaptively switches between two modes based on per-step prediction confidence. When confidence is high (above threshold τ), SOAR decodes all high-confidence positions in parallel to reduce latency; when confidence is low, it falls back to Position Beam Search (PBS)—a beam search over alternative unmasking orderings in the position space—to avoid premature commitments. PBS is introduced as the core technical contribution: unlike standard AR beam search that explores the token/vocabulary space, PBS explores which positions to unmask at each step, generating beam candidates by selecting different subsets of masked positions. SOAR dynamically adjusts the beam size after each step based on whether the best candidate originated from parallel decoding (collapse to 1) or PBS (expand to K). Experiments on GSM8K, MBPP, and HumanEval with DREAM-7B-Base and LLaDA-8B-Base show accuracy improvements of up to +5.5% (HumanEval on DREAM) while maintaining competitive or faster inference speed compared to greedy decoding.

**Compliance With Llm Reviewing Policy:**

Affirmed.

**Key Questions For Authors:**

1. What is the performance on instruction-tuned models (e.g., DREAM-7B-Instruct, LLaDA-8B-Instruct)? These are more practically relevant, and their confidence distributions may differ from base models.

2. How does SOAR compare to best-of-N sampling with greedy decoding at equivalent total compute (N forward passes)? For example, generating 2 greedy samples and selecting the one with higher average confidence would use similar compute to SOAR with K=2 but without the position search overhead.

**Limitations:**

The authors should discuss potential failure modes of confidence-based switching more explicitly—e.g., cases where high confidence is miscalibrated, or where the threshold τ requires task-specific tuning despite the claimed robustness. The absence of instruction-tuned model evaluations is a practical limitation that should be acknowledged.

**Strengths And Weaknesses:**

Strengths:
1. The paper is clearly written and well-organized. Figure 2 provides an effective visual overview of the SOAR algorithm. The Pareto frontier analysis (Figure 3) nicely illustrates the quality-speed tradeoff landscape across methods. The progression from PBS (quality but slow) to PBS+Parallel (fast but quality drop) to SOAR (balanced) tells a coherent story.
2. The ablation studies are thorough: confidence threshold sensitivity (Figure 4b-c), beam size (Figure 5), alternative unmask metrics (margin, negative entropy in Table 2), and compatibility with variable-length decoding via DreamOn (Table 3). The AR-ness analysis (Section 4.4, Figure 6b) provides an interesting secondary insight: SOAR tends to produce less left-to-right-biased unmasking schedules, which correlates with better accuracy.

Weaknesses:
1. The confidence-based adaptive parallel decoding component is acknowledged as being "similar to the design of Fast-dLLM (Wu et al., 2025)" (Section 3.3.1). The novel component—PBS—is essentially standard beam search transplanted to the position dimension. While the application is new, the algorithmic contribution is limited. The combination via confidence switching, while practical, is a straightforward engineering decision.
2. The paper evaluates only on Base models (DREAM-7B-Base, LLaDA-8B-Base). Instruction-tuned models, which are more relevant for practical deployment, may have different confidence distributions and different sensitivity to unmasking order. The absence of instruct-model experiments limits the practical conclusions.

---

> ### Author Rebuttal · Authors · 2026-03-31
>
> **Response to Reviewer hkNQ**
>
> We sincerely thank the reviewer for the detailed and constructive feedback. We have carefully considered the comments and address them below.
>
> ---
>
> >**W1. Novelty and Empirical Insights**
>
> We highlight several aspects that distinguish SOAR beyond a simple combination of existing ideas.
>
> - **Algorithmic novelty.** SOAR is a training-free framework that dynamically switches between position beam search and parallel decoding based on per-token confidence, a non-trivial integration unexplored in prior DLM inference work.
>
> - **Quality without sacrificing speed.** SOAR achieves consistent quality gains while maintaining competitive or superior inference speed.
>
> - **Novel empirical insights.** Beyond exploiting the known correlation between confidence and correctness, our work further confirms it: actively searching for higher-confidence decoding paths leads to measurable accuracy improvements, demonstrating that confidence is an actionable signal rather than a mere byproduct. We additionally uncover a negative correlation between Global AR-ness and quality, offering new directions for understanding DLM decoding behavior.
>
>
>
> ---
>
> >**Q1. Evaluation on Instruction-Tuned Model**
>
> We compare the confidence distributions of the Base model and the Instruct model on LLaDA (Table 2‑1), using greedy decoding on GSM8K.
>
> Table 2‑1: Confidence distributions of Base model and Instruct model
>
> | Model | [0, 0.8) | [0.8, 0.995) | [0.995, 1.0] |
> |-------|:-:|:-:|:-:|
> | LLaDA-8B-Base | 34.4% | 17.4% | 48.2% |
> | LLaDA-8B-Instruct | 10.9% | 28.7% | 60.4% |
> | Difference | −23.5% | +11.3% | +12.2% |
>
>
> We observe that the **Instruct model is more confident** in its output tokens. To assess behavioral differences under this higher-confidence condition, we conducted supplementary experiments on LLaDA-8B-Instruct using the same threshold as for the Base models, and provide results for both **Pure Diffusion** and **Block Diffusion** settings.
>
> As shown in Table 2‑2, SOAR achieves accuracy improvements without sacrificing speed in most settings.
>
> Table 2‑2: Results on LLaDA-8B-Instruct
>
> | LLaDA-8B-Instruct | HumanEval (256) | HumanEval (512) | MBPP (256) | MBPP (512) | GSM8K (256) | GSM8K (512) |
> |---|:-:|:-:|:-:|:-:|:-:|:-:|
> | *Pure Diffusion* | | | | | | |
> | Greedy | 37.2 | 45.7 | 39.0 | 40.0 | 63.5 | 68.9 |
> | +SOAR | 40.3 | 45.7 | 39.8 | 41.4 | 65.4 | 70.3 |
> | SpeedUp | ×2.73 | ×2.09 | ×1.31 | ×1.08 | ×1.80 | ×2.11 |
> | *Block Diffusion* | | | | | | |
> | Greedy | 37.8 | 48.8 | 34.4 | 36.6 | 77.6 | 77.5 |
> | +SOAR | 39.6 | 46.9 | 39.2 | 39.4 | 78.1 | 78.5 |
> | SpeedUp | ×2.47 | ×2.07 | ×1.04 | ×1.09 | ×2.04 | ×2.63 |
>
>
> ---
>
> >**Q2. Comparison with Best-of-N Sampling**
>
> We thank the reviewer for suggesting best-of-N as a baseline. To ensure diversity for best-of-N, we use temperature = 0.5 and report results averaged over 3 runs on HumanEval (max_len=256, block=32).
>
> We compare:
> - Greedy Decoding
> - best-of-N (N=2): generate 2 samples, select the one with higher average confidence
> - Position Search (Ours): beam_size=2 (equivalent compute)
> - SOAR (Ours): search only when confidence is low, with beam_size=2
>
> As shown in Table 2‑3, under equivalent computation, Position Search achieves higher Pass@1 than best-of-N. We attribute this to that Position Search seeks higher-confidence paths at every step, resulting in higher average confidence.
>
> Additionally, SOAR performs search only when confidence is low, and parallel decodes when confident. Consequently, its computational cost is significantly lower than both Position Search and best-of-N, and can be >2× faster than greedy.
>
> Table 2‑3: Comparison with best-of-N sampling
>
> | | Greedy | Best-of-N | Position Search | SOAR |
> |---|:-:|:-:|:-:|:-:|
> | Pass@1 | 37.4±2.3 | 37.8±3.4 | 38.7±0.5 | 38.6±2.4 |
> | Time (min) | 23.2±2.6 | 46.3±2.6 | 45.6±1.3 | **9.1±0.1** |
> | Avg Conf. | 0.956±0.000 | 0.961±0.001 | 0.965±0.001 | 0.963±0.001 |
>
>
> >**Limitations**
>
> We thank the reviewer for this suggestion. We agree that failure modes of confidence-based switching deserve explicit discussion.
>
> **Locally confident but globally incorrect branches.** SOAR can favor incorrect paths when they contain more locally confident tokens, yielding higher average confidence at the decision point. This reflects a limitation of token-level confidence: it optimizes for local rather than global sequence quality, and can be misled by short-term confidence spikes.
>
> **Over-confident models.** On RL-optimized models (e.g., justGRPO [1]), most tokens have confidence near 1, making it difficult for SOAR to trigger search. This suggests robustness may degrade on miscalibrated models.
>
> We will add a Limitations section in the camera-ready version discussing these issues and span-level confidence estimation as a future direction.
>
> ---
>
> >**References**
>
> [1] Ni, Zanlin, et al. "The Flexibility Trap: Why Arbitrary Order Limits Reasoning Potential in Diffusion Language Models." *arXiv preprint arXiv:2601.15165* (2026).

---

> > ### Author Rebuttal · Reviewer_hkNQ · 2026-04-06
> >
> > The authors performed new experiments on missing instruct model and also included the required baseline, I will consider raise the score, currently I am a bit on the fence of it.

---

> > > ### Author Response · Authors · 2026-04-06
> > >
> > > Thank you for the follow-up. We are glad that the additional experiments helped clarify SOAR’s behavior on instruction-tuned models and its benefit relative to a compute-matched best-of-N baseline. We will incorporate these results, together with the expanded limitations discussion, into the final version.
> > >
> > > We hope these clarifications and additional evidence will be taken into account in your final assessment, and we sincerely appreciate your thoughtful consideration.
> > >
> > > Sincerely,
> > >
> > > The Authors

---

### Official Review · Reviewer_aEHj · 2026-03-19

**Soundness:** 2
**Presentation:** 3
**Significance:** 2
**Originality:** 2
**Overall Recommendation:** 4
**Confidence:** 2

**Summary:**

The paper proposes a training-free decoding method for diffusion language models. It argues that standard greedy decoding is limited because the choice of which positions to unmask is itself a search problem. To address this, the authors introduce Position Beam Search (PBS), which explores multiple candidate unmasking trajectories, and SOAR, an adaptive method that switches between search and faster parallel decoding based on model confidence.

Experiments on math and code benchmarks with DREAM-7B and LLADA-8B show that PBS can improve generation quality, while SOAR aims to retain much of that gain with better decoding efficiency.

**Compliance With Llm Reviewing Policy:**

Affirmed.

**Final Justification:**

My concerns were addressed in rebuttal. Raising my score to 4.

**Key Questions For Authors:**

How does SOAR compare to the closest prior and concurrent decoding-order search methods under a matched evaluation setup?

**Limitations:**

yes

**Strengths And Weaknesses:**

Strengths
1. The paper addresses a practically relevant problem for diffusion language models by improving the quality-efficiency tradeoff at inference time without requiring retraining.
2. The proposed methods are technically clear and well specified. PBS and SOAR are described concretely, and the overall approach is methodologically appropriate for the paper’s goal.
3. The paper is generally well written and easy to follow. The motivation is clear, and the progression from greedy decoding to PBS and then to SOAR is coherent.
4. The empirical evaluation is reasonably aligned with the claims, covering two DLM families and multiple math and code benchmarks, along with useful ablations.

Weaknesses
1. The originality appears moderate, as the paper mainly combines search-based decoding and confidence-based adaptive acceleration, both of which are already active directions in related inference literature.

2. The related-work positioning could be sharper in clarifying what is genuinely new relative to prior adaptive decoding and decoding-order search methods.

---

> ### Author Rebuttal · Authors · 2026-03-31
>
> **Response to Reviewer aEHj**
>
> We sincerely thank the reviewer for the thoughtful and constructive feedback. We have carefully considered the comments and would like to address them in detail below.
>
> ---
>
> >**W1&2. Originality of SOAR**
>
> We would like to highlight several aspects that distinguish SOAR from a simple combination of existing ideas.
>
> - **Algorithmic novelty.** SOAR is a training-free framework that dynamically switches between position beam search and parallel decoding based on per-token confidence, a non-trivial integration unexplored in prior DLM inference work.
>
> - **Quality improvement without sacrificing speed.** Unlike pure search methods that improve quality at the cost of significant slowdown, SOAR achieves consistent quality gains while maintaining competitive or superior inference speed compared to greedy decoding.
>
> - **Novel empirical insights.** Beyond exploiting the known correlation between confidence and correctness, our work further confirms it: actively searching for higher-confidence decoding paths leads to measurable accuracy improvements, demonstrating that confidence is an actionable signal rather than a mere byproduct. We additionally uncover a negative correlation between Global AR-ness and quality, offering new directions for understanding DLM decoding behavior.
>
> We hope these points clarify the contributions of SOAR and appreciate the reviewer's constructive feedback.
>
> ---
>
> >**Q1. Comparison with Concurrent Work**
>
> Regarding the positioning of our work relative to prior adaptive decoding and decoding-order search methods, we acknowledge that the related-work discussion could be sharper. In the revised version, we will include additional comparative experiments to further clarify the advantages of our approach.
>
> For decoding-order search methods, to the best of our knowledge, **Order-Token Search** [1] is the most related concurrent work. We would like to clarify that we did not draw on this work throughout the entire project, and this work was not published on arXiv before the ICML submission. Order-Token Search achieves strong general improvements by searching across both position and token dimensions, but suffers from higher time complexity. The novelty of our method lies in: (1) performing search only when the model's confidence is low, avoiding excessive computation on high-confidence tokens; (2) employing parallel decoding only when the model's confidence is high. We demonstrate that flexibly switching between the two modes based on the model's own confidence can simultaneously improve both decoding quality and decoding speed. We will incorporate these clarifications into the related work section in the next version.
>
> Since the most related work Order-Token Search did not release its code, it is difficult to conduct a comparison under perfectly matched settings. Therefore, we compare against the results reported in their paper using the same benchmarks (HumanEval), the same maximum lengths (256, 512), the same decoding approach (Block Diffusion with block=32), and the same model LLaDA-8B-Instruct, based on Opencompass.
>
> Additionally, we also compare with **FourierSampler** [2], a method that guides decoding positions based on frequency, using the results reported in their paper under the same experimental settings.
>
> As shown in Table 1-1, SOAR performs better on HumanEval. We would like to emphasize that SOAR only performs search when the model exhibits low confidence, and combines it with parallel decoding when confidence is high. This not only avoids the inference latency introduced by search, but also achieves a 2×+ inference speedup across both decoding lengths. This is consistent with the motivation of our method: improving inference quality without sacrificing inference speed.
>
> ---
>
> Table 1-1: Comparison with concurrent methods on HumanEval. * indicates that the speedup compared to greedy decoding is estimated based on the theoretical complexity of the method.
>
> | Method | HumanEval (256) | HumanEval (512) |
> |--------|:-:|:-:|
> | Order-Token Search [1] | 34.2 | 37.2 |
> | SpeedUp | ×0.5* | ×0.5* |
> | FourierSampler [2] | — | 40.9 |
> | SpeedUp | — | ×1.0* |
> | SOAR (Ours) | 39.6 | 46.9 |
> | SpeedUp | ×2.47 | ×2.07 |
>
> We hope this clarifies the contribution of SOAR and may help the reviewer reconsider its novelty. We sincerely appreciate the reviewer’s time and thoughtful feedback.
>
> ---
>
> >**References**
>
> [1] Shen, Yangyi, et al. "Improving Diffusion Language Model Decoding through Joint Search in Generation Order and Token Space." *arXiv preprint arXiv:2601.20339* (2026).
>
> [2] He, Siyang, et al. "FourierSampler: Unlocking Non-Autoregressive Potential in Diffusion Language Models via Frequency-Guided Generation." *arXiv preprint arXiv:2601.23182* (2026).

---

> > ### Author Rebuttal · Reviewer_aEHj · 2026-04-06
> >
> > My concerns were addressed. I'll raise my score.

---

> > > ### Author Response · Authors · 2026-04-06
> > >
> > > Thank you very much for your positive feedback and for taking the time to review our work. We greatly appreciate your recognition and are glad that the additional experiment addressed your concerns. Your support is very important to us.
> > >
> > > Sincerely,
> > >
> > > The Authors

---

### Decision · Program_Chairs · 2026-04-30

**Decision:**

Accept (regular)

**Comment:**

There are initially mixed reviews for this submission. Reviewers had concerns regarding originality (aEHj), positioning among related work (aEHj, HW4Z, T5Gd), and lack of evaluation on instruct models (hkNQ), which was sufficiently addressed during rebuttal. After rebuttal, all reviewers are leaning towards acceptance. Therefore, I'm recommending acceptance conditioned on the authors integrating the results they provided during rebuttal into their final version.